# Qualitative Data Clustering to Detect Outliers

**DOI:** 10.3390/e23070869

**Published:** 2021-07-07

**Authors:** Agnieszka Nowak-Brzezińska, Weronika Łazarz

**Affiliations:** Institute of Computer Science, Faculty of Science and Technology, University of Silesia, Bankowa 12, 40-007 Katowice, Poland; weronika.lazarz@smcebi.edu.pl

**Keywords:** qualitative data, outliers detection, data clustering, *K-modes*, *ROCK*, *STIRR*

## Abstract

Detecting outliers is a widely studied problem in many disciplines, including statistics, data mining, and machine learning. All anomaly detection activities are aimed at identifying cases of unusual behavior compared to most observations. There are many methods to deal with this issue, which are applicable depending on the size of the data set, the way it is stored, and the type of attributes and their values. Most of them focus on traditional datasets with a large number of quantitative attributes. The multitude of solutions related to detecting outliers in quantitative sets, a large and still has a small number of research solutions is a problem detecting outliers in data containing only qualitative variables. This article was designed to compare three different categorical data clustering algorithms: *K*-modes algorithm taken from MacQueen’s *K*-means algorithm and the STIRR and ROCK algorithms. The comparison concerned the method of dividing the set into clusters and, in particular, the outliers detected by algorithms. During the research, the authors analyzed the clusters detected by the indicated algorithms, using several datasets that differ in terms of the number of objects and variables. They have conducted experiments on the parameters of the algorithms. The presented study made it possible to check whether the algorithms similarly detect outliers in the data and how much they depend on individual parameters and parameters of the set, such as the number of variables, tuples, and categories of a qualitative variable.

## 1. Introduction

The basis of effective machine learning of intelligent systems is an effective data analysis based on the insights of qualified analysts – experts in their specialization. Data analysts typically have advanced technical skills that machines cannot cope with, such as identifying the right questions, finding the right data sources, and interpreting the results. Most of all, creating analytical data sets from which the intelligent system will develop its own analysis schema. Thus, data analysis enables an effective viewing of steadily growing datasets which are being continously generated by both companies and customers. It also provides quick information and recommendations based on access to real-time information. Data analysis methods focus on strategic approaches to extracting raw data, retrieving information relevant to the company’s core goals, and searching for that information in order to turn metrics, facts and figures into tools that bring improvement. The selection of data analysis methods depends on the type of data analyzed. The methods for quantitative data are different from those for qualitative data. Data need preparation. For this purpose, the data are checked for any outliers, missing values, or the necessary data discretization. One of the biggest problems in business reporting is to combine and view data of different types with different characteristics. With data analysis, one can search many different types of data for correlations and insights. The process of “mixed” research, incorporating quantitative and qualitative research techniques, methods, approaches, and concepts together into one research, was introduced in the 1970s by Denzin [1].

Quantitative research focuses on collecting quantitative data and generalizing it into groups or explaining a specific observation. Another type of data is qualitative data that exists when you can make descriptive statements about a topic based on observations, interviews or evaluation.

In this article, we dealt with the detection of unusual data in qualitative data. This topic is challenging. While we can easily to determine whether numerical data represents a deviation, this task is very complex in case of qualitative data. For example, the following two features describe a human being: height in centimeters and eye color expressed as text. Let us also assume that in the set used for comparitive purpose, we will select 3 people: A, B, and C. Person A is 185 cm tall and has blue eyes, as is the vast majority of data in the set (let’s say 90%). Person B is 148 cm tall with brown eyes - one of three in the entire dataset. Finally, person C is 178 cm tall (exactly what the average height for all people in the data set is) and has black eyes (the only one in the whole set). Without much difficulty, we will find (based on the calculations for the numerical attribute of height) that person B is a deviation here - because the 148 cm value differs significantly from the average of 178 cm. Detecting deviations for qualitative characteristics, such as eye color, is a much more complex task. Because we cannot say unequivocally whether black or brown is more different than blue from the information stored as text, all we can do is conclude that black was the rarest occurrence, so this is a potential deviation.

It is worth explaining for what purpose we detect outliers. We do not assume at all that they must appear in a given set. In our research work so far, we deal with data clustering algorithms for data mining and faster searching. It is known that after dividing the data into groups, we create representatives for each group. Then in the search process, these representatives are analyzed when we look for a group that matches the analyzed data. Thus, if there are outliers in the dataset, and we use a clustering algorithm that is not immune to noise in the data, unfortunately, these outliers will negatively affect the quality of the clusters and thus the quality of the search. We want our solution to be universal and, therefore, to work effectively both for data with outliers and for typical data. We also want our solution to be effective for any data type, not just only numerical, which is easier to analyze [2,3,4].

We assume that by finding outliers in a dataset, we can draw the attention of an expert in the field, who has a chance to check whether an outlier is an error or maybe a real data, unlike the rest, and worth a deeper analysis. In case of medical data, these may be unusual disease symptoms, which will allow experts in the field to explore them more widely and get to know the topic better. If there are no outliers in the set, the clustering should proceed without disturbances, and thus searching for clusters, or their representatives, should provide the expected results.

In our research, we often analyze qualitative data. We obtain such data directly from domain experts, e.g., in the form of rule-based knowledge bases, where rules are created using qualitative attributes. We can also generate them from numerical data through the discretization process. This is due to the fact that much more useful information can be derived from qualitative rather than quantitative data. Going back to the aforenmetioned example of height, we realize that the information about the average value for this feature does not mean all people in the set are more or less of that average height. We would have to support ourselves with information, e.g., about the median and standard deviation, to find out more. However, when we consider the feature of eye color, if we know that the most common eye color is blue (we got a dominant value), it gives us much more key information. When browsing a large dataset, we often formulate questions without giving specific values, e.g., for height, we do not provide the specific value we are looking for but only approximate the search, stating that we are looking for tall people. Since qualitative data is such a frequently used representation of data, we want to check how the clustering algorithms deal with this data and whether it is possible to detect unusual values in such data sets during clustering.

Therefore, we chose completely random sets of qualitative data in our experiments. We do not assume that there are outliers to be found. So we want to check how the three selected clustering algorithms: *K*-modes, ROCK and STIRR deal with detecting deviations if they occur.

Clustering data or combining data into clusters means dividing a data set into subsets, including elements similar to each other in terms of a certain selected measure.

An outlier can be defined as a data object that appears to be inconsistent with the rest of the dataset based on some measure, or as an observation that deviates significantly from other observations giving rise to suspicions about the correctness of the data or the behavior of the system. Frank E. Grubbs in [5] writes, “An outlying observation, or outlier, is one that appears to deviate markedly from other members of the sample in which it occurs.” Then, he describes the different types of outliers: outliers, which are a symptom of noise in the data and have valuable information, so they should be treated as other objects in the set, and cases that are errors that we must remove from the set. Such objects or observations often contain useful information about the abnormal behavior of the system described by the data, and remaining undetected could lead to incorrect model specification, erroneous parameter estimates, and incorrect results. Data cleansing requires invalid data to be identified and handled appropriately. Outliers that are detected are often candidates for invalid data that we should remove from the set. Often, in order to extract useful knowledge from large data sets, experts use data mining techniques. One of the frequently used techniques is, for example, clustering algorithms. When such algorithms work on imperfect data, i.e., data containing outliers, we should expect that they will affect the clustering results. The resulting clusters will be distorted and may be difficult to analyze and interpret later. For this reason, the research related to the detection of outliers in the structures of data clusters is certainly important and will improve the quality of data mining in such sets.

The question that may be asked is why we chose these three algorithms. The answer is quite simple. First, we have chosen them because we had previously paid a lot of attention to clustering algorithms. We know that the most popular algorithms are the *K*-means - a partitional algorithm, and the Agnes algorithm which represents hierarchical algorithms for numerical data. The two algorithms selected by us, *K*-modes and ROCK, respectively, are equivalents of the algorithms indicated here for qualitative data. We found the STIRR algorithm particularly interesting, and since we did not find a Python implementation for it, we decided to fill the gap.

We want to compare whether the selected algorithms: *K*-modes, ROCK and STIRR indicate the same objects as potential deviations. To do this, we look for 5%, 10%, and 15% variations and check the coverage of the results. We can see that despite the analysis of as many as nine different data sets, we cannot state clearly which methods are more similar to each other, i.e., they indicate similar outliers.

## 2. State-Of-The-Art

Outlier detection methods are used by IT specialists on a daily basis in many systems supporting the organization of everyday life, such as detecting credit card fraud, clinical trials, analysis of voting irregularities, data cleansing, detecting network intrusions, predicting difficult weather conditions, providing geographic information, athlete performance analysis and much more. Different applications of outlier detection require the use of case-specific methods. In many applications, outliers are more interesting than internal values [6]. Fraud detection is a classic example where the focus is on outliers because in cases of fraud they can provide more information than the data in the norm. The outliers detected may indicate individuals or groups of customers whose behavior is outside what is considered the norm. In article [7], the variance isolation method, MI−Local−DIFFI (Multiple Indicator Local-DIFFI), was used to identify bank customers who have unusual transactions that indicate money laundering. The authors have used a modified Isolation Forest algorithm by assigning weights to individual features during the construction of isolation trees [8]. Early detection of financial negligence within financial institutions prevents corruption, terrorism and the imposition of penalties on institutions, and helps to eliminate cases of bank customer insolvency. The authors of [9] present advanced anomaly detection systems in network traffic through various categorizations and conduct a comparative study of Dropbox, Google Drive and iCloud security measures with qualitative data about internet users.

Detection of outliers is also used to support mechanisms that ensure the flow of information, support for logistics, medicine and economy. Stored records of patients with unusual symptoms or test results can help detect potential health problems in an individual patient. The identification of such abnormal observations will help to distinguish recording errors from whether the patient fell on potential diseases, and thus to take effective medical measures in time. In the article [10] the DBSCAN algorithm was used to clear the dataset from unusual observations. The prepared data was then transferred to the Random Forest model alerting patients to the risk of diabetes and hypertension in their early stages. The authors of [11] present a method of graphical presentation of qualitative demographic and personal data of patients to detect ouliters among hospitals that treat patients with myocardial infarction. Likewise, logistics related to the management and control of the flow of products from the source of production to the destination requires the cleaning of potential anomalies from the data. It is very important to ensure the safety and reliability of the goods during this process. Transportation tracking can help spot potential outliers such as product quality and quantity or goods damage. The article work in [12] describes the use of the LOF (Local Outlier Factor) algorithm to detect traffic outliers that are data errors or unusual traffic events in everyday situations such as accidents, traffic jams, and very little traffic. In their work on dynamic process monitoring Yuxin Ma et al. used the LOF algorithm to monitor unexpected changes in operating conditions in order to quickly adapt to the prevailing conditions on the basis of a CSTR tank reactor [13]. The cited examples of well-known and novel methods for detecting data outliers are based on quantitative or mixed sets. So far, few methods using categorical data have been described, and most of them were in the form of individual interviews or manual analysis using simple models, e.g., linear and graphical presentation [14].

The methods of detecting outliers in datasets can be divided into formal and informal. Most formal tests require test statistics to test hypotheses. They usually rely on the assumption of some well-behaved distribution and check whether the extreme target value is out of range, e.g. extremely high air temperature in winter. Although formal tests are quite powerful with well-behaved statistical assumptions, most real-world data distributions may be unknown or may not follow specific distributions. There may also be an anomaly masking problem where one anomaly is hidden by another, next to which it appears to be a normal value. Apart from the masking problem, there is the problem of swamping, in which normal observations are treated as anomalies because they are too close to outliers [15]. Therefore, the anomaly detection method depends on the distribution and type of data as well as the knowledge about the set. The topic of anomaly detection is still being developed and described with the help of many alternative solutions combining known and popular methods with innovative research.

Cluster-based approaches can effectively identify outliers as points that do not belong to dataset clusters or as clusters distinguished by a small number of features [16,17]. The methods for determining such values based on distance are applied based on a measure of the distance between the object and its closest neighbors in the dataset. They have better computational performance than depth-based methods, especially in large datasets, and are effective in identifying local outliers in datasets with multiple clusters. The methods based on neural networks can independently model the distribution of input data and distinguish appropriate and inappropriate classes [18]. Those data objects that do not reproduce well in the output layer are considered outliers. Such methods effectively identify outliers and automatically reduce activation functions based on key attributes. Support vector machine (SVM) methods can distinguish between good classes and outliers by mapping data into a feature space [19]. Objects distant from most of other objects or located in relatively sparse regions of the feature space are declared as outliers. These methods effectively identify outliers for quantitative data for which there is no need to prepare them in advance. Model-based methods detect outliers by building a model that can represent the statistical behavior of datasets or of the objects themselves [20,21].

Outliers are objects that differ significantly from the learned model. These methods can deal effectively with various types of data and are often used in detecting outliers in an online data stream or the aforementioned recurring banking transactions.

So far, a lot of papers have been published focusing on detecting outliers and good data clusters in quantitative and mixed sets with binary categorical variables. M. Breunig, in [22], proposed a method for detecting local outliers. According to its assumptions, a data object is assigned the value of the local outlier factor (LOF), which is calculated from the ratio of the local density of that object and the local density of its nearest neighbors in the number of NPts. The objects with the highest LOF values are considered outliers. One method that works well for small size datasets is the minimum size ellipsoid technique described in 1984 by P. J. Rousseeuw in the article [23], which uses the smallest allowable size of an ellipsoid to define a boundary around most data. Objects are outliers if they are not in a densely populated boundary. A popular algorithm for processing large datasets to detect outliers is the DBSCAN proposed by Martin Ester et al. described in [24] and its modified version of ADBSCAN described a little later in [25]. The basic condition for using the algorithm is to know two parameters: ϵ and MinPts. Epsilon (ϵ) is the radius of the closest neighborhood of the data object under consideration, and MinPts is the minimum number of data objects present in its region. Based on these parameters, the algorithm creates clusters in the data. Objects isolated from the rest are considered outliers. Another model-based method that isolates outliers instead of normal objects is IsolationForest . The method is based on the construction of a forest of binary isolation trees in the number of which is a model parameter. Then outliers are observations with the average shortest path lengths from the root of the tree to the leaf. The algorithm was described in the article [8]. The indicated algorithms are widely used in IT systems, both to clean data sets from noise so that they do not interfere with the system operation, and to detect unusual observations in the data for further analysis.

The presence of outliers in qualitative data can significantly disrupt the effectiveness of machine learning algorithms that try to find patterns in the data, such as rules, for example, decision rules or association rules. Having two data objects, one of which is an outlier, differs only in the dependent variable. We can get an unwanted rule determined by the outlier. To eliminate problems such as these, algorithms are created to divide the dataset into clusters understood as dense regions in the attribute space. Intuitively, the cluster consists of much more tuples than expected when all the attributes are independent. Moreover, the cluster also extends to the largest possible region. The division of the dataset into clusters storing similar objects concerning a certain measure makes it possible to eliminate outliers stored in very small clusters. So far, very few algorithms for processing this type of data have been described. The five most popular methods so far to perform this task are the following algorithms: *K*-modes, CLOPE, STIRR, ROCK and CACTUS. Each of the algorithms represents a completely different approach to clustering categorical data.

The CLOPE algorithm was first proposed by Yiling Yang et al. in 2002 [26]. The authors propose to cluster qualitative data using histograms for the frequency of occurrence of qualitative features. High rates indicate typical data. Low rates may suggest variations in the data. The CACTUS algorithm built based on the STIRR algorithm presents the most advanced and computationally complex approach to data clustering [27]. CACTUS consists of three phases: summarization, clustering, and validation. In the summary phase, the similarities within the feature and between the features are calculated. In the clustering phase, sets of candidate clusters are discovered based on the summary information. The last validation phase consists of selecting the final set of clusters from among the candidate clusters based on the set threshold of objects included in the clusters.

Based on the examples provided, it can be seen how important a part of any data analysis is detection of outliers is, not only to eliminate them so that they do not cause problems in further analysis, but also to draw conclusions and the possibility of reacting to unusual behavior of systems or tested objects. They also show the need for an in-depth analysis of the methods of detecting anomalies in datasets that cannot be described numerically or more simply expressed in a qualitative form. So far, no papers describing the application of the indicated algorithms on a large scale or comparing the results due to the type of data processed and the time of execution have been found. This has become the direct motivation of the authors of this paper to analyze three selected clustering algorithms *K*-modes, STIRR and ROCK that allow to effectively detect outliers in the qualitative data.

## 3. Data Clustering

The problem of clustering is one of the most studied issues in social sciences, psychology, medicine, machine learning and data science.

The basic classification of clustering methods includes partitioning-based methods, hierarchical methods, density-based and depth-based methods. Partitioning clustering is a clustering method used to classify cases in a dataset into multiple groups based on their similarity. The algorithms require the analyst to determine the number of clusters to generate. These algorithms minimize the given clustering criterion by moving data objects between clusters until optimal clustering is achieved. Examples of partitioning algorithms include the K−means method described in [28] or the CLARA algorithm described in [29]. A separate group of algorithms that divides data into clusters is hierarchical algorithms. These methods are designed to create groups of data that are significantly different from each other, and the objects within the groups are similar to each other. Hierarchical clustering begins with categorizing each object as a separate group. Then it repeatedly finds two clusters that are most similar to each other and merges them into one. The process ends when no pair of clusters can be joined because they are too far apart from some selected measure. Representatives of a group of hierarchical algorithms can be the ROCK algorithm described in [30], which will be presented in detail in this section. Density-based algorithms are based on the assumption that the data cluster is a contiguous region, which is dense due to the selected distance measure. Two objects belong to the same cluster if their distance from each other does not exceed this measure and the cluster density is greater than a certain value. Objects belonging to regions with low density (with few objects belonging to a cluster) are usually considered outliers. The DBSCAN method, presented for the first time in 1995 and described in [22], is well known and most associated with the density-based methods of data clustering. In addition to density-based clustering, a similar approach is presented by the depth-based algorithms described in [31]. Popular methods of data clustering also include fuzzy methods, methods using statistical models, and machine learning models [32,33,34,35]. All data clustering methods have one thing in common—they divide the dataset into subsets of objects similar to each other. They are commonly used to separate normal observations from noise in the data.

In addition to typical benefits of clustering data, it has found a wide application in the processing of datasets with categorical (or qualitative) domains, both in the process of preparation for mining and in the modeling process itself. Here, data clustering has been used to find outliers in qualitative datasets. The three methods described in this chapter differ in terms of data clustering and outliers detection. Neither algorithms directly defines the variances in the datasets but clusters the objects to determine those with the greatest distance from the rest (they form single-element clusters or are the farthest from the center of the clusters). The *K*-modes algorithm, most often used in research and real IT systems, creates groups of clusters from objects closest to selected centroids and defines outliers as objects farthest from the cluster center. A similar technique is used by the STIRR algorithm, which does not directly define data clusters. Still, it assigns each object a similarity coefficient to the rest of the set and naturally forms the dataset’s central object. Outliers are the features with the coefficient farthest from zero. The last of the described algorithms, the ROCK algorithm, calculates the similarity measures between objects and between groups of objects, creating data clusters containing objects that should not belong to any other cluster. The easiest way to identify outliers in clusters created this way is to select single-element clusters.

### 3.1. K-modes Clustering

The *K*-modes clustering algorithm was first proposed and made public in [36] by Z. Huang in 1997 as an alternative to the categorical data of the popular *K*-means algorithm. The modifications made to the *K*-means algorithm include using a simple measure of matching dissimilarity for qualitative features, replacing the group averages with vectors composed of the most common values at individual coordinates of the objects (modes), and using a frequency-based method to modes update.

The *K*-modes algorithm begins with a random selection of *k* objects (centroids) selected by the user, which are the central objects of *k* clusters. Then, the dissimilarity measure is calculated for each pair of the drawn object and each other object in the dataset, and the closest centroid is determined for each object. When all objects are assigned to individual clusters, the centroids are then updated by creating new modes from objects present in the cluster. The calculations are repeated until the differences in the generated clusters in the following steps cease to exist.

What is important in this algorithm is that while we randomly select *k* modes from the dataset, for each pair (mode, object) the dissimilarity measure is calculated. For each object that is not a mode, we find the mode closest to the object. Then, we join objects with the corresponding modes to create clusters.

As with most of categorical clusters, clusters containing a tiny number of features or a single feature can be considered outliers. The specifics of *K*-modes clustering show that we will create single-element clusters only if the initially drawn object is an outlier. That is, it shows the least similarity with the rest of the objects.

Finding the similarity between a data object and a cluster requires *n* operations, which for all *k* clusters is nk. Assigning objects to the appropriate *k* clusters and updating modes also require nk operations. Assuming the algorithm is run *I* times for different starting objects, the algorithm will have a linear complexity of O(nkI).

The *K*-modes algorithm is the easiest to implement and the most popular among the aforementioned categorical data clustering algorithms because it is linearly scalable concerning the size of the dataset, and it is easy to implement and handles large categories of data efficiently. The disadvantage of the algorithm is that it selects random initial modes, leading to unique structures clustering around objects that are undesirable from the set. A method to prevent such situations to some extent is to draw the initial set of modes multiple times and assign each object to the cluster with the greatest number of times. The output clusters generated by the *K*-modes algorithm have a similar cardinality, which does not have to reflect the actual data clusters on the sets having atypical distributions of variables.

### 3.2. STIRR Clustering

STIRR (Sieving Through Iterated Relational Reinforcement), a dynamic system described by D. Gibson, is one of the most influential methods of clustering qualitative datasets [37]. It represents each attribute value as a weighted vertex in a graph. It iteratively assigns and propagates weights until a fixed point is reached. Different weight groups correspond to different clusters on the attribute.

Assuming that the input set is composed of *n* tuples containing only qualitative data in the number of *m* variables, the described algorithm will create a separate node from each unique variable value. The unique value here is the value of a categorical variable. This concept refers to a single value of a given variable, not to the entire subset of records with that value. For example, for a variable with the coordinates [a,a,b,b,b], the unique values are *a* and *b*. *m* nodes will identify each tuple. We will refer to the configuration as assigning a weight to each of the *v* nodes. For obvious reasons, the weights must add up to the number 1.

There are many methods of selecting the initial weights of nodes, the most common of which is to adjust the weight to the occurrence of values - the more occurrences, the higher the weight will be. We can also set the weights of all values to 1 or randomly select weights from the [0;1] range and then normalize them. The authors of the algorithm propose to use the Gram–Schmidt orthonormalization method as a function that normalizes weights [38].

The computational complexity will depend on the similarity of the objects and the selected number of iterations. Assuming *n* is the number of objects in the dataset, *d* is the number of unique values in all variables, and *k* is the number of user-defined iterations, the computational complexity of the algorithm will be O(kdn). If a large *k* is selected and there is no repeatability in the successively calculated weights, the order of complexity may increase to $O(dn^{2})$.

The STIRR method largely depends on the choice of parameters - the joining operator and the iterative function that defines the dynamical system. The algorithm ends when each unique value of the variable receives a positive or negative weight so that reporting end clusters can involve a heavy postprocessing step. In addition, by selecting different configurations of the initial weights, the set will be divided in different ways, which concludes that the initial weight has a big impact on the final result. Defining outliers as those objects whose sums of weights on individual coordinates are close to zero may turn out to be sufficient, assuming that the set consists of two clusters. As the assigned positive and negative weights indicate a high coexistence of values, STIRR quickly identifies values that do not coexist with any other values and can be considered last. Due to the specificity of the algorithm consisting in consulting unique values in a set, STIRR is very efficient in detecting outliers in even huge datasets, the variables of which have a small variety of occurring values. The computation time of the algorithm increases with increasing unique values, especially if the input set contains continuous scale variables.

### 3.3. ROCK Clustering

ROCK (Robust Clustering Algorithm for Categorical Attributes) was first put forward in *Information Systems* by S. Guha et al. as a hierarchical algorithm for categorical data proposing an approach based on a new concept called summaries between data objects [30]. The algorithm helps overcome the difficulties of applying Euclidean measures over multivariate vectors with categorical values. An important element in selecting the appropriate clusters is the determining of the function of the criterion. The best object clusterings are those that gave the highest values. To find clustering that maximizes the criterion function, a measure of goodness is used that determines the best cluster pair to merge at each stage of the ROCK hierarchical clustering algorithm. The algorithm considers the neighbourhood of individual pairs of objects. It starts by assigning each tuple to a separated cluster, and then clusters are merged repeatedly according to the closeness between clusters. The closeness between clusters is defined as the sum of the number of ”links” between all pairs of tuples, where the number of ”links” is computed as the number of common neighbors between two tuples. Let us define $link(x_{1},x_{2})$ to be the number of common neighbors between $x_{1}$ and $x_{2}$. From the definition of links, it follows that if $link(x_{1},x_{2})$ is large, then it is more probable that $x_{1}$ and $x_{2}$ belong to the same cluster. Since our goal is to find a clustering that maximizes the criterion function, we use a measure similar to the criterion function in order to determine the best pair of clusters to merge at each step of ROCK’s hierarchical clustering algorithm. For a pair of clusters $C_{i}$, $C_{j}$, let $link[C_{i},C_{j}]$ store the number of cross links between clusters $C_{i}$ and $C_{j}$, that is, $\sum_{x_{1}\in C_{i},x_{2}\in C_{j}}^{}link\left(x_{1},x_{2}\right)$. Then, we define the goodness measure $g(C_{i},C_{j})$ for merging clusters $C_{i}$, $C_{j}$.

ROCK’s hierarchical clustering algorithm accepts as input the set *X* of *n* sampled objects to be clustered (that are drawn randomly from the original dataset), and the number of desired clusters *k*. The procedure begins by computing the number of links between pairs of objects. Initially, each object is a separate cluster. For each cluster *i*, we build a local heap q[i] and maintain the heap during the execution of the algorithm. q[i] contains every cluster *j* such that link[i,j] is non-zero. The clusters *j* in q[i] are ordered in the decreasing order of the goodness measure with respect to *i*, g(i,j). In addition to the local heaps q[i] for each cluster *i*, the algorithm also maintains an additional global heap *Q* that contains all the clusters. Furthermore, the clusters in *Q* are ordered in the decreasing order of their best goodness measures. Thus, g(j,max(q[j])) is used to order the various clusters *j* in *Q*, where max(q[j]), the max element in q[j], is the best cluster to merge with cluster *j*. At each step, the max cluster *j* in *Q* and the max cluster in q[j] are the best pair of clusters to be merged. The while-loop iterates until only *k* clusters remain in the global heap *Q*. In addition, it also stops clustering if the number of links between every pair of the remaining clusters becomes zero. In each step of the while-loop, the max cluster *u* is extracted from *Q* by extract max and q[u] is used to determine the best cluster *v* for it. Since clusters *u* and *v* will be merged, entries for *u* and *v* are no longer required and can be deleted from *Q*. Clusters *u* and *v* are then merged to create a cluster *w* containing |u| + |v| objects. The details are described in [30]. The ROCK algorithm can successfully identify outliers that are relatively isolated from the rest of the objects. Objects with very few or no neighbors in clusters of one or several members will be counted as outliers.

Due to a large number of comparisons between objects to create the best-defined clusters, the algorithm’s computational complexity is the greatest of all those described in this paper. Assuming that $m_{m}$ is the maximum number of neighbors and $m_{\alpha }$ is the average number of neighbors, the computational complexity of the algorithm is $O(n^{2}+nm_{m}m_{\alpha }+n^{2}\log n)$. The overall computational complexity will depend on the number of neighbors of each facility. In most cases, the order of complexity will be $O(n^{2}\log n)$. If the maximum and an average number of neighbors are close to *n*, then the algorithm’s complexity increases to the order of $O(n^{3})$. ROCK belongs to the family of hierarchical algorithms. It is unique because it assumes that an attribute value, in addition to its frequency, must be examined based on the number of other attribute values with which it occurs. Due to its high computational complexity, ROCK is good at detecting deviations in small datasets, and its computational time increases as the records in the set increase. This is because each record must be seen as a unique data cluster. Data sampling has a huge impact on the algorithm’s results, which can be considered its basic disadvantage. If the user does not have comprehensive knowledge about the dataset, the appropriate selection of the θ value and the minimum number of clusters generated on the output is a challenging task.

## 4. Data Description

We have used nine qualitative datasets to compare the algorithms that detect deviations in the data, each with a different structure of the variables matched to the clustering-based algorithms supporting the detection of deviations in the qualitative datasets. The sets have different sizes and consist of a different number of categorical explanatory variables with a strong influence on the dependent variable. For a reliable representation of the time complexity of the algorithms and the method of data classification in terms of deviated observations, we have selected sets with a different number of objects and variables and a different number of classes of the qualitative variable.

### 4.1. Primary Tumor Dataset

The Primary Tumor Dataset is one of three databases provided by the Institute of Oncology, which has appeared many times in the literature on machine learning. The collection was obtained from the University Medical Center of the Institute of Oncology in Ljubljana and published by M. Zwitter and M. Soklic [39]. The collection concerns primary tumors in humans. Locations of primary tumors are places in the body where the tumor appeared for the first time, and from there began to form new tumors in other parts of the body. Data objects are characterized by patient variables such as age, gender, skin type, and sites of metastasize. The purpose of the dataset analysis is to determine the starting point at which the tumor appeared. The set contains 338 records, 18 columns and 49 unique values.

### 4.2. Lymphography Dataset

The dataset is part of primary cancer research in humans. Like the Primary Tumor Dataset, the Lymphography dataset was obtained from the University Medical Center of the Institute of Oncology in Ljubljana, and published by M. Zwitter and M. Soklic [40]. The set contains 148 records, 19 columns and 62 unique values.

### 4.3. Congressional Voting Records Dataset

The voting dataset in the US Congress was designated by Jeff Schlimmer [41]. This is the result of the 1984 United States Congress vote. Each record corresponds to one congressman’s vote on 16 issues (e.g., education expenditure, crime). All attributes are logical values. The dataset contains records for 168 Republicans and 267 Democrats. The set contains 434 records, 17 columns and 34 unique values.

### 4.4. Car Evaluation Dataset

The car rating database was derived from a simple hierarchical decision model originally developed for the DEX demonstration by M. Bohanec, V. Rajkovic [42]. The car rating database directly ties a car to the six input attributes: purchase, maintenance, doors, persons, boot space, safety. Due to the known concept structure, this database can be beneficial for testing constructive induction methods and structure discovery. The set contains 1728 records, 7 columns and 26 unique values.

### 4.5. SPECT Heart Dataset

The dataset describes the diagnosis of single-proton emission computed tomography cardiac images. Each patient falls into two categories: normal and abnormal. The database contains 267 records and 23 binary attributes. The collection was published on the UCI Machine Learning platform by Krzysztof J. Cios and Lukasz A. Kurgan [43].

### 4.6. Effects on personality due to Covid-19

The dataset describes changes in people’s behavior and feelings during the Covid 19 epidemic. Data were likely obtained from individual interviews. The dataset comes from the Kaggle platform. The collection contains 204 records described with 17 categorical attributes [44].

### 4.7. Phishing website Detector Dataset

The collection contains websites described by 30 parameters and a class label identifying them as a phishing sites. Phishing sites aim to obtain personal information by pretending to be trusted organizations. The collection was published on the Kaggle platform [45]. The original dataset contains over 11,000 records but was reduced to 1842 records to simplify the calculation. The set describes 31 variables. Some data have been discretized to achieve an entirely qualitative dataset.

### 4.8. Japanese Credit Screening Dataset

The examples represent positive and negative cases of people who have been granted or refused a loan. The collection was generated from interviews with people in a Japanese loan company. The collection was published on the Data World platform by Chiharu Sano [46]. The collection consists of 690 objects and 16 attributes. Some data have been discretized in order to achieve an entirely qualitative dataset.

### 4.9. Bank Data for Cash Deposit

The dataset describes the attributes of Bank customers that could encourage them to open a long-term deposit. The collection includes data such as age, occupation, education, and credit history. The collection was published on the Kaggle platform [47]. The original dataset contains over thirty thousands of records, but has been reduced to 2539 records for ease of computation. Some data have been discredited in order to achieve an entirely qualitative dataset.

Preparation of datasets required filling the missing values with the most common value in the variable. As objects contain text values on coordinates, the time complexity of the algorithms increases significantly when there is a comparison of the values of two objects due to the need to compare each letter in the text. In order to speed up the operation of the algorithms, a function encoding text into numerical values has been used, which are seen by the algorithms as qualitative variables. Variable codes are stored in a dictionary structure, so we can easily decode them after the algorithm returns the result.

## 5. Conducted Research

The algorithms described in Section 3 have been implemented in the Python language and tested on the sets described in Section 4. The JupyterHub environment available at the address https://jupyter.org/hub (accessed on April 2021) was used for the implementation and visualization of the data. JupyterHub runs in the cloud or on hardware locally and supports a preconfigured data science environment for each user. Installation of the environment requires the prior installation of Python on your own hardware. In this work, we have used the Python language version 3.6 and the Anaconda package containing most of the libraries enabling the execution of machine learning models and data mining and visualization of results. The existing models of the Scikit−Learn library have been used to implement the *K*-modes algorithm. due to the lack of implementation, the algorithms ROCK, STIRR were implemented by the authors. We have used the Matplotlib library and the PandasDataframe structure for data visualization. Most of the computation is based on the Pandas data structures that hold the results. The diagram (Figure 1) shows the sequence of steps performed to compare the algorithms detecting outliers.

The computer program described by the authors has been embedded in the JupyterHub environment, which enables the separate execution of subroutines without the need to run the entire program code. This type of program management is used in exceptionally large and computationally time-consuming programs, because it enables multiple code testing without the need to compile it completely, and is useful in data visualization “on the fly”. The program has been divided into sections containing the following:Import of Python analytical libraries SciPy,Scikit−learn,NumPy,Pandas,Matplotlib and libraries to perform operations related to time.Implemented algorithms: STIRR (stirr) with threshold parameter denoting the percentage of expected outliers, ROCK (rock) with parameters: *k* denoting the expected number of clusters and theta being a parameter of a function that returns an estimated number of neighbors between objects in clusters and *K*-modes (k_modes) with *k* parameter denoting the expected number of clusters and threshold parameter denoting the percentage of expected outliers.Data preparation functions: (fill_empty) – function that completes missing fields with the most common value in a column and removes columns that contain more than 60% empty values, (variable_coding) function encoding text values into numerical values, taking a list of columns whose values will be encoded and returning an encoded dataset and a dictionary enabling data decoding, an analogous function decoding encoded text variables (variable_encoding).Uploading all datasets.Reads the selected Car
Evaluation
Dataset and lists several records.Calculation of descriptive statistics for the Car
Evaluation
Dataset.Encoding text variables for the selected Car
Evaluation
Dataset to visualize the result.Loading and preparation of other datasets.Execution of STIRR, ROCK and *K*-modes algorithms on datasets. Presentation of the graph of time dependence on the choice of the algorithm and the input set.Presentation of graphs of the dependence of the execution time on the number of variables, the number of records and data diversity.Listing the numbers of individual clusters obtained by the ROCK, *K*-modes algorithms and the numbers of the two resulting clusters obtained by the STIRR algorithm.Listing the weights assigned to the values in the STIRR algorithm based on the Car
Evaluation
Dataset to present the results.Showing the Car
Evaluation
Dataset with assigned cluster numbers for the ROCK and *K*-modes algorithms, sums of weights for each coordinate for the STIRR algorithm and for all algorithms, flags that indicate whether a record has been classified as an outlier. If the flag is −1, the object is an outlier, if it is 1, the object is considered normal.Presentation of the matrix of similarities and differences in classifying values as outliers for the STIRR, ROCK and *K*-modes algorithms compared in pairs.Identification of common outliers generated by the STIRR, ROCK, and *K*-modes algorithms.

Full analysis of the results will be discussed in the next section.

The source of the software was placed in the GitHub repository https://github.com/wlazarz/outliers (accessed on May 2021). The repository contains the implementation of three clustering algorithms for data that has only quality domains: *K*-modes, STIRR (Sieving Through Iterated Relational Reinforcement) and ROCK (Robust Clustering Algorithm for Categorical Attributes).

## 6. The Results of the Experiments

This section covers the results of the three algorithms described in the previous sections: STIRR, ROCK, and K−modes. An important element of data preprocessing was their proper preparation, which is described in the Section 6.1. Then, the algorithms were compared in terms of their time complexity and conclusions were drawn about the desired structures of the input datasets for individual algorithms. In the following, the authors describe the process of generating clusters and determining outliers together with the visualization of results, and summarize the deprecated deviations.

### 6.1. Data Preprocessing

The first step in the project was to load datasets and prepare them properly before starting clustering. In all datasets, we completed empty fields with the most common value on a given variable. Categorical variables were encoded into numeric variables on *Primary Tumor Dataset*, *Lymphography Dataset*, *Congressional Voting Records Dataset* and *Car Evaluation Dataset*. Despite reducing the dataset to a numerical form, algorithms working on qualitative sets treat numbers as categories of variables. The process of numerically encoding the test values was intended to reduce the long execution time of the algorithms resulting from the need to compare each sign of the test value. Table 1 and Table 2 show the encoding of Car Evaluation Dataset.

### 6.2. Time Complexities of Clustering Algorithms

Based on the sets described in Section 4, we performed an analysis of the time complexity of the algorithms described in the work. The execution time of the algorithms is given in seconds. The study was conducted in the JupyterHub environment installed locally on MacBookPro hardware with IntelCorei7 quad-core processor and 16 GB RAM. As datasets are characterized by a different number of objects and variables and represent different types of data, the number of records, columns, and avoided values for each dataset is described on the lower axis to facilitate the analysis of time complexity (Figure 2).

It is easy to notice that in most cases, the STIRR algorithm worked the longest, which justifies its very high computational complexity, and it is extremely dependent on the number of *d* denoting the number of unique values for individual variables. Simple calculations show that unfortunatelly $O(dn^{2})$ complexity of STIRR (in pesimistic case) is higher than square logarithmic $O(n^{2}+nm_{m}m_{\alpha }+n^{2}\log n)$ computational complexity of the ROCK algorithm (assuming that $m_{m}$ is the maximum number of neighbors and $m_{\alpha }$ is the average number of neighbors in the cluster) in cases where *d* is greater than $1+\frac{m_{\alpha }m_{m}}{n}+\log n$, which is presumably the case for the first three sets. In these cases, the number of unique values in the set is at least 8% of the multiple of the record set. We can observe another situation by analyzing the time complexity of the algorithms on the car dataset. In this case, the time complexity of the ROCK algorithm is several times higher than the time complexity of the STIRR algorithm. This effect can be caused by a small number of unique values (26) compared to a large number of records in the dataset. As an estimate, if the $m_{m}$ and $m_{\alpha }$ values are greater than 9% of the set, then *d* will reduce the computational complexity of the STIRR algorithm. It results from the equation $26<1+\frac{m_{\alpha }m_{m}}{1728}+\log1728$ which holds if $m_{m}m_{\alpha }>24336$. In the case of the ROCK algorithm, the computational complexity increases with the size of the dataset, in contrast to the STIRR algorithm, whose complexity increases with the number of unique values in the set. This behavior of the algorithm is intuitive, knowing that the STIRR algorithm focuses on a hypergraph that stores unique values of individual categorical variables. The *K*-modes algorithm has an average linear or near-square complexity when diagnosed with many clusters. It is the least complex of the three described algorithms, which is also reflected in Figure 2. Regardless of the number of records, variables, and values, the execution time for the *K*-modes algorithm is the lowest for each dataset. More detailed information on the ratio of the time complexity of the algorithm to the number of data and the values of the variables can be found on the set of four graphs (Figure 3).

We can observe that the complexity of the ROCK algorithm increases very quickly with the increase in amount of data (here, the product of the number of records and variables). In contrast, the time complexity of the STIRR algorithm increases with the increase of unique values for individual variables, which will confirm the previous considerations.

### 6.3. Detected Clusters

Algorithms working on qualitative datasets required the indication of individual parameters for the dataset. In the case of the *K*-modes algorithm, the parameter was the number of generated clusters. The ROCK algorithm parameters were the minimum number of generated clusters and the parameter of the function returning the estimated number of neighbors between objects in the clusters. Implementing the ROCK algorithm became a tough challenge due to a very high computational complexity and unusual parameters. The algorithm was initially tested on a dataset containing 13,000 records. To properly select the θ parameter and the final number of clusters, the algorithm was tested for all θ from the [0.5;0.95] range in steps of 0.01 and the number of clusters in the [2;80] range. Due to the need to calculate all connections between the objects, the algorithm initially had to perform 170 million operations. The test lasted over a day. Ultimately, we selected the nine smaller sets described in Section 4 to present the results. We selected the ROCK algorithm parameters on a trial and error basis. The number of outliers was approximately 5% of the set, assuming that single-member clusters would be classified as outliers and stored in one common cluster with the 0 number. To reliably test the differences in the results of the ROCK, STIRR and *K*-modes algorithms, the threshold of outliers for the last two algorithms was also set to a number corresponding to the number of outliers generated by the ROCK algorithm (e.g., if the ROCK algorithm-generated five single-element clusters, the other two algorithms will return five outliers in the dataset). The division of the dataset into clusters performed by all three algorithms and the number of outliers found is shown in Figure 4, Figure 5 and Figure 6, assuming that the algorithm parameters are selected individually for the dataset.

While the Robust
Clustering (ROCK) algorithm analyzes the similarities not only between objects, but also between clusters that should be merged into a single cluster, the STIRR and *K*-modes algorithms arrange objects from a dataset between clusters so that each cluster contains a similar amount of data and only focuses on the similarities between individual objects in the data. As mentioned earlier, the definition of an outlier generated by the ROCK algorithm, taken from S. Guha, R. Rastogi and K. Shim, indicates one-element classes. The more difficult task was to mark the anomalies using the other two algorithms based on qualitative data. For this reason, the detection of outliers by these two algorithms was performed separately after we completed cluster generation. As suggested by the authors, in the case of the STIRR algorithm, the records with the sums of weights on individual coordinates close to zero were considered as outliers. The records marked as anomalies by the *K*-modes algorithm are the ones from the farthest neighborhood of the centroid in which the cluster the object is located.

All datasets used in work were taken from the *UCI Machine Learning Repository* database and represent real data collected during research on real data objects with different distributions, possibly containing a small number of deviations, which results in differentiated sizes of clusters generated by the ROCK algorithm. This is not the case for the other two algorithms due to their implementation; algorithms distribute objects between clusters so that the number of objects in the clusters is similar. The descriptions of the clusters and the anomalies detected by the three indicated algorithms (Figure 4, Figure 5 and Figure 6) show that the set with the strongest links between the objects is the *Car Evaluation Dataset*, for which all algorithms indicated less than 1% outliers. When analyzing the distribution of variables in the car set (Figure 7), we can observe fluctuations only in the safety variable. All datasets processed by the ROCK algorithm have created a single, power-dominant cluster containing most of the records in the set. Such a mapping of sets in clusters results from numerous connections between groups of records that are naturally similar to each other.

The STIRR algorithm is unique because of its implementation concerning the rest of the algorithms for clustering qualitative data. The algorithm determines the weight for each unique value in a set of qualitative data concerning the number of connections between the value and other values that coexist in the set. The determined numerical weights are normalized with the Gram–Schmidt orthonormalization to reach values in the range [−1;+1]. In a further step, the algorithm calculates the weight of each data object, which is the sum of the weights of the values on the individual coordinates of the object. Object weights naturally divide the dataset into two clusters: a cluster containing objects whose sum of weights is a negative number and a cluster that contains objects whose sum of weights is a positive number. As mentioned, the outliers of the user-selected number, in this case, are the records whose sums of weights are closest to zero. Figure 8 shows the weights assigned to the unique values of each variable of the *Car Evaluation Dataset*, presented in the form of a double-key dictionary, where the first key coordinate is the variable’s value and the second is the variable’s name. In this case, the middle value of *(“acc”, ”safety”)* has the lowest weight among the values of the set variables, which means that it has the least coexistence with the values of other variables.

Data clustering algorithms do not have natural definitions of outliers and do not return points considered as variances in the data. The problem of marking objects that differ the most from the others due to the calculations characteristic of the algorithm was solved by generating an additional column for the dataset containing the values −1 or 1, where −1 means that the object was considered an outlier and 1 means that we knew the object to be normal. Table 3 shows a part of the output dataset *Car Evaluation Dataset* and the cluster numbers generated by the ROCK and *K*-modes algorithms, with the sum of weights on each coordinate generated by the STIRR algorithm and values of 1 or −1 assigned to objects.

### 6.4. Detected Outliers

In order to compare the algorithms for detecting outliers based on clustering, objects classified as outliers and normal were found by each couple of pairs of algorithms (Table 4).

It is easy to see that we can only find a common outlier in the *house votes* dataset. In most of the cases, the analyzed algorithms returned completely different results. The large differences in the selection of outliers are certainly the result of a significantly different nature of cluster detection by each of the algorithms. The ROCK algorithm is the most rigorous in detecting outliers. It focuses on inter-object and inter-cluster connections, tying them together until well-defined clusters are obtained with the number of common neighbors below a certain threshold. Thus, single-member clusters contain distant objects from every other cluster and every data object. The STIRR method represents a completely different approach.

The acronym stands for *Sieving Through Iterated Relational Reinforcement* and provides an easy explanation for method of iterating through relationships between unique values in a dataset until the best set of weights assigned to values is discovered. Outliers determined by the STIRR algorithm are objects whose values on individual variables show little coexistence with other values in the dataset. The last algorithm, K−modes, due to the inoculated randomness during the selection of the initial set of cluster centroids, will consider those outliers whose distance from the centroids in the clusters to which they belong is the greatest. Due to a very different approach to determining good clusters, and therefore detecting outliers by the three described algorithms, the anomaly classification result will also be different for each algorithm.

Due to the lack of correlations between the classification results, the experiment was performed assuming that the algorithms should generate 10% of outliers. The results of this study are shown in Table 5.

This time the result is more satisfactory. Appropriate operations were performed to read that the number of objects marked as anomalies by all three algorithms was two. An intersection operation was performed to determine the objects identified as anomalies in the dataset by all three algorithms. This time, we found common outliers in the car (Table 6), house votes (Table 7), and japanese credit (Table 8), spect (Table 9), alpha bank (Table 10), phishing (Table 11) datasets.

When analyzing Table 11, we can see that the outliers are not characterized by the very high absolute weights that were calculated by STIRR algorithm, but they are large enough to be considered as anomalies. We also found these objects in the ROCK single-element clusters and 10% of the objects farthest from the cluster centroids generated by the K−modes algorithm. Thus, with little risk, the anomaly detection result can be based on commonly detected outliers by several algorithms, assuming that the percentage of outliers in the set is greater than it should be, and then select objects that have been classified as anomalies by all the algorithms. Then we can be sure that the set has been tested in terms of differences between the attributes, attribute groups, objects, and groups of objects. Therefore, we can design the anomaly search process in a qualitative set in two steps. Initially, all algorithms for the small anomaly threshold can search for common anomalies. If the process does not return results, one can increase the threshold and see if there are common outliers in the set this time. If the threshold exceeds approximately 15% of the set, we can find no outliers.

## 7. Evaluation of the Proposed Methods

The analysis of the results allows us to conclude that if we care about the speed of calculations or have a huge set of data, a good choice will be to use the *K*-modes algorithm. This algorithm is recommended to be used in datasets that we know are divided into a small number of large clusters. Then the initially drawn centroids will have less influence on the quality of clustering.

When we know that the dataset contains obvious outliers, i.e., those that differ by most of the coordinate values from the rest of the objects, it is worth using the STIRR algorithm. This algorithm looks for relationships between the values of individual variables. Still, it does not pay much attention to whether a given value repeatedly occurs, which could indicate the existence of a specific data cluster based on a small number of variables.

In most cases, the most reasonable approach is to use the ROCK method because it performs an exhaustive analysis of the dataset in search of outliers – it approaches objects, variables individually, and looks for relationships between objects and variables. The main disadvantage of this algorithm is a very high computational complexity, which in extreme cases may amount to the cube value of the number of objects in the set. For this reason, the algorithm is a good choice if we have small datasets, up to 1000 records. An additional difficulty is the selection of the distance between the clusters and the minimum number of clusters. The algorithm’s execution time and the clustering quality are improved by knowing an estimated number of clusters in the set and how far the elements should be apart from each other to not be in a common cluster. If we do not have exhaustive knowledge about the dataset, it is worth running the algorithm many times and analyzing the generated clusters to assess the quality of the parameters.

## 8. Conclusions

This work focused on the search for outliers in sets of qualitative data due to the type and number of variables. Section 3 describes relatively novel approaches to clustering qualitative data. The results are presented on the basis of nine datasets characterized by a different structure. In the multitude of solutions related to the clustering of quantitative data, clustering of data containing only qualitative variables are large and still have a small number of research solutions. Research into anomaly detection and impact observations has placed emphasis on quantitative datasets, and the methods of working with large categorical databases have been left to a small number of alternative techniques.

The authors have attempted to compare the effectiveness of cluster detection and anomaly detection in qualitative datasets, between which there is no clear comparison so far. The algorithms based on quantitative data generally have better mathematical properties enabling a simple data visualization, which may be sufficient to match a technique to the dataset we have. This does not apply to qualitative datasets, so determining which algorithm works better on the data we have and detecting the most natural groups is difficult. The performance of algorithms is usually defined in terms of their scalability and cluster generation time. If the dataset under analysis changes each day and the daily data clustering technique is part of the data mining process, any changes to the data will affect the results. Then extreme variations in daily data are undesirable and will disrupt cluster generation. The stability of qualitative clustering algorithms is an insufficiently researched issue and still raises a lot of interest in the world of Data Science.

We can draw a basic conclusion from the research that the data structure has a large impact on the time complexity of the algorithm. The algorithm should be properly selected for the dataset. Each algorithm classifies outliers differently, and the results will differ from one another. With a high probability, the objects classified as outliers by each algorithm can be considered noise in the set and removed before further exploration on the set. The algorithms based on clustering categorical data are relatively new methods of detecting outliers in data, having no implementation in commonly used programming languages. The quality of the created clusters is measured by the user’s knowledge and the examination of the results because the user sets the basic parameters of clustering, which require a very large knowledge of the data. The clustering techniques used in categorical data are based on the coexistence of data objects or the number of their neighbors and are therefore not suitable for detecting clusters in quantified or mixed data. The discussed STIRR, ROCK, and K−modes algorithms introduce different methods to solve this problem and provide different solutions in terms of their performance with respect to the time needed to execute the algorithms when the number of records and dimensions change.

## Figures and Tables

**Figure 1 entropy-23-00869-f001:**
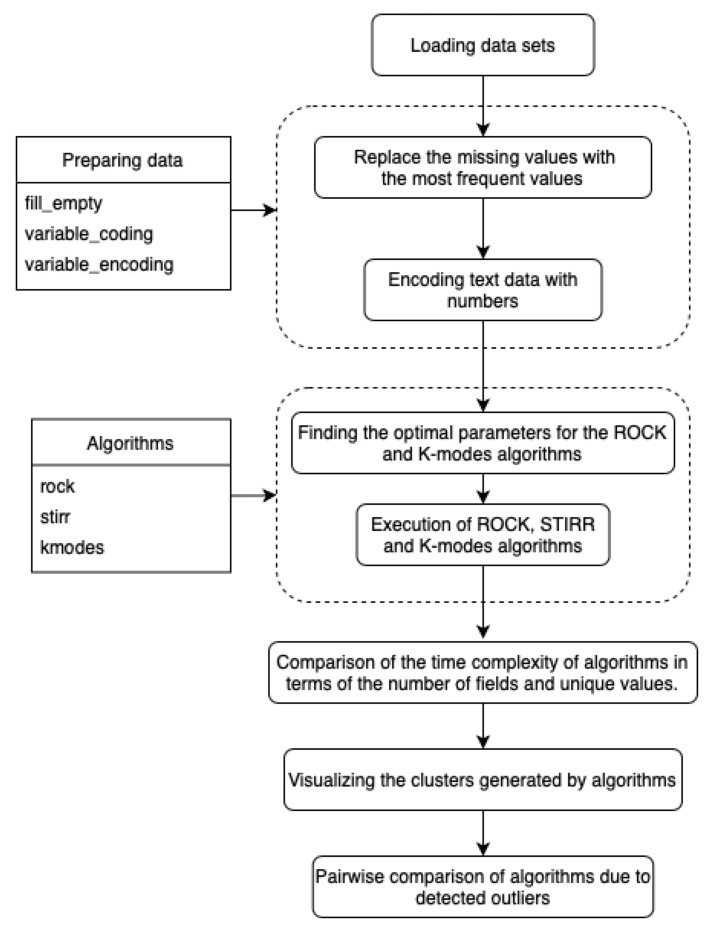
Scheme of the program comparing algorithms clustering data and detecting outliers.

**Figure 2 entropy-23-00869-f002:**
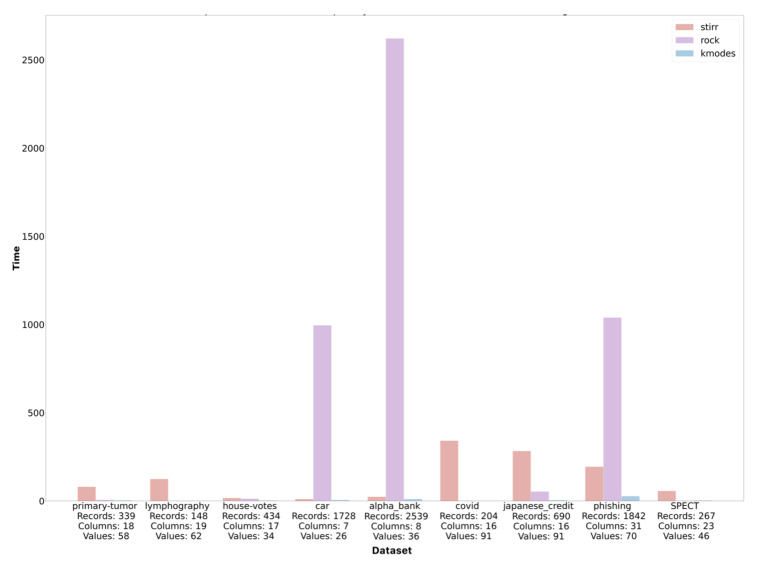
Comparison of the time complexity of the STIRR, ROCK and K−modes algorithms.

**Figure 3 entropy-23-00869-f003:**
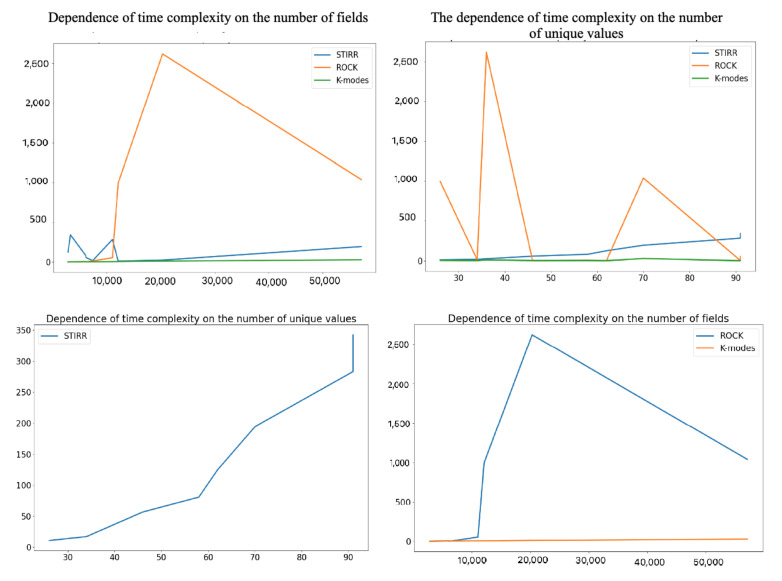
Dependence of the time complexity of the STIRR, ROCK, and K−modes algorithms on the number of records and unique values.

**Figure 4 entropy-23-00869-f004:**
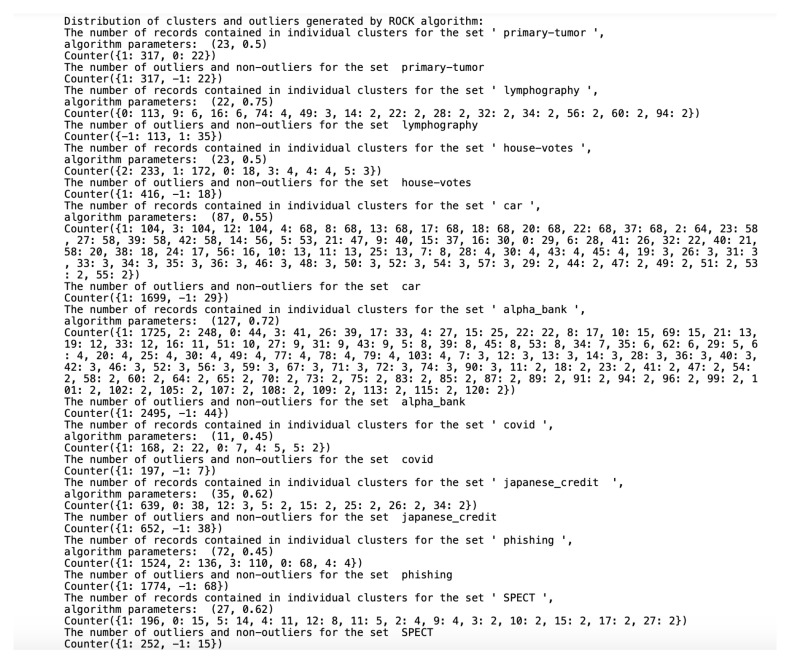
Distribution of clusters and outliers generated by ROCK.

**Figure 5 entropy-23-00869-f005:**
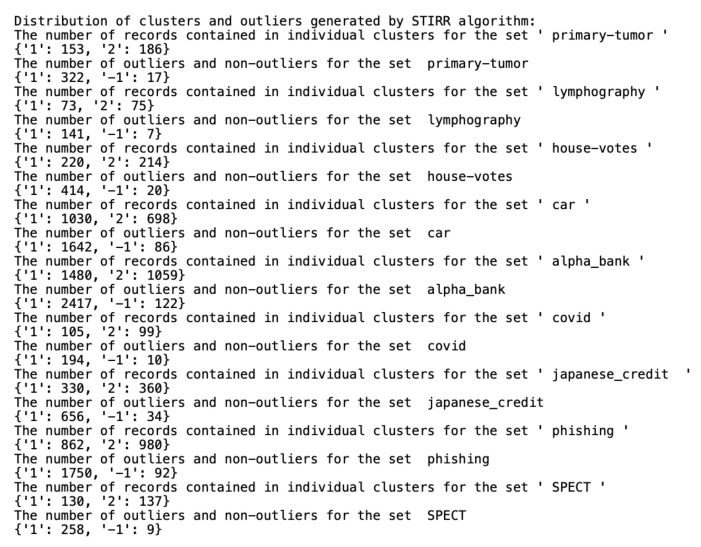
Distribution of clusters and outliers generated by STIRR.

**Figure 6 entropy-23-00869-f006:**
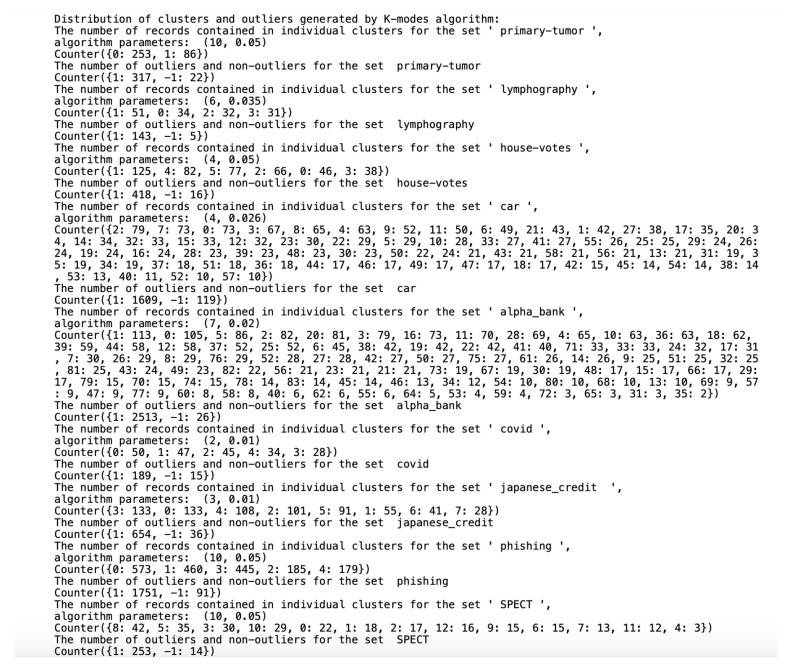
Distribution of clusters and outliers generated by *K*-modes.

**Figure 7 entropy-23-00869-f007:**
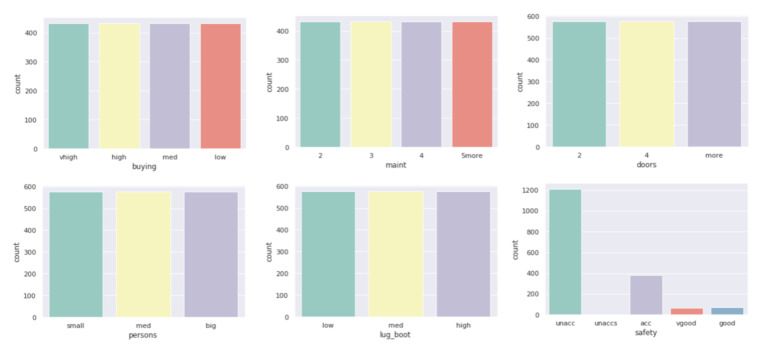
Variable histograms in Car Evaluation Dataset.

**Figure 8 entropy-23-00869-f008:**
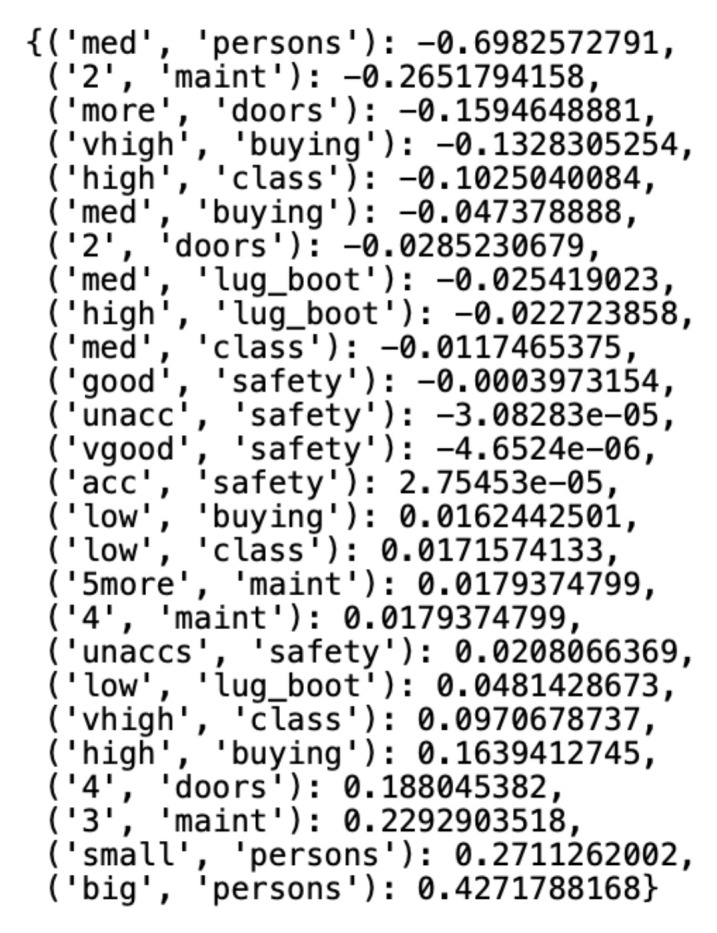
Variable class weights generated by the STIRR algorithm.

**Table 1 entropy-23-00869-t001:** Part of Car Evaluation Dataset before encoding categorical variables.

Class	Buying	Maint	Doors	Persons	Lug_BOOT	Safety
vhigh	vhigh	2	2	small	low	unacc
vhigh	vhigh	2	2	small	med	unacc
vhigh	vhigh	2	2	small	high	unacc
vhigh	vhigh	2	2	med	low	unacc
vhigh	vhigh	2	2	med	med	unacc
vhigh	vhigh	2	2	med	high	unacc
vhigh	vhigh	2	2	big	low	unacc
vhigh	vhigh	2	2	big	med	unaccs
vhigh	vhigh	2	2	big	high	unacc
vhigh	vhigh	2	4	small	low	unacc

**Table 2 entropy-23-00869-t002:** Part of Car Evaluation Dataset after encoding categorical variables.

Class	Buying	Maint	Doors	Persons	Lug_BOOT	Safety
4	8	12	15	16	20	22
4	8	12	15	16	21	22
4	8	12	15	16	19	22
4	8	12	15	17	20	22
4	8	12	15	17	21	22
4	8	12	15	17	19	22
4	8	12	15	18	20	22
4	8	12	15	18	21	26
4	8	12	15	18	19	22
4	8	12	15	16	20	22

**Table 3 entropy-23-00869-t003:** Part of *Car Evaluation Dataset* with cluster numbers and flags for outliers or normal values.

	CLASS	Buying	Maint	Doors	Persons	Lug_BOOT	Safety	ROCK Clusters	STIRR Weights	*K-modes* Clusters	ROCK Labels	STIRR Labels	*K-modes* Labels
0	vhigh	vhigh	2	2	small	low	unacc	1	−0.010227	1	1	−1	1
1	vhigh	vhigh	2	2	small	med	unacc	1	−0.083789	1	1	1	1
2	vhigh	vhigh	2	2	small	high	unacc	1	−0.081094	1	1	1	1
3	vhigh	vhigh	2	2	med	low	unacc	1	−0.97961	1	1	1	1
4	vhigh	vhigh	2	2	med	med	unacc	1	−1,053172	1	1	1	1
...	...	...	...	...	...	...	...	...	...	...	...	...	...
1723	low	low	5 more	more	med	med	good	32	−0.832199	0	1	1	−1
1724	low	low	5 more	more	med	high	vgood	41	−0.829112	0	1	1	1
1725	low	low	5 more	more	big	low	unacc	0	0.367165	0	−1	1	1
1726	low	low	5 more	more	big	med	good	32	0.293237	0	1	1	1
1727	low	low	5 more	more	big	high	vgood	41	0.296325	0	1	1	1

**Table 4 entropy-23-00869-t004:** Comparison of commonly classified outliers and normal values by the STIRR, ROCK and K−modes algorithms assuming 5% outliers.

Comparison	STIRR&ROCK	STIRR&*K-modes*	ROCK&*K-modes*	Dataset
−1 and −1	0	0	17	primary-tumor
1 and 1	300	300	312	primary-tumor
different result	39	39	10	primary-tumor
−1 and −1	6	0	5	lymphography
1 and 1	34	136	35	lymphography
different result	108	12	108	lymphography
−1 and −1	2	1	7	house-votes
1 and 1	398	399	407	house-votes
different result	34	34	20	house-votes
−1 and −1	0	1	2	car
1 and 1	1613	1524	1582	car
different result	115	203	144	car
−1 and −1	3	2	8	alpha_bank
1 and 1	2376	2393	2477	alpha_bank
different result	160	144	54	alpha_bank
−1 and −1	0	1	3	covid
1 and 1	187	180	185	covid
different result	17	23	16	covid
−1 and −1	1	1	19	japanese_credit
1 and 1	619	621	635	japanese_credit
different result	70	68	36	japanese_credit
−1 and −1	2	6	52	phishing
1 and 1	1684	1665	1735	phishing
different result	156	171	55	phishing
−1 and −1	0	1	5	SPECT
1 and 1	243	245	243	SPECT
different result	24	21	19	SPECT

**Table 5 entropy-23-00869-t005:** Comparison of commonly classified outliers and normal values by the STIRR, ROCK and K−modes algorithms assuming 10% outliers.

Comparison	STIRR&ROCK	STIRR&*K-modes*	ROCK&*K-modes*	Dataset
−1 and −1	4	1	1	primary-tumor
1 and 1	254	284	263	primary-tumor
different result	81	54	75	primary-tumor
−1 and −1	3	1	1	lymphography
1 and 1	116	120	115	lymphography
different result	29	27	32	lymphography
−1 and −1	7	5	22	house-votes
1 and 1	367	360	389	house-votes
different result	60	69	23	house-votes
−1 and −1	16	12	30	car
1 and 1	1370	1335	1325	car
different result	342	381	373	car
−1 and −1	30	43	249	alpha_bank
1 and 1	1904	1807	1842	alpha_bank
different result	605	689	448	alpha_bank
−1 and −1	1	2	6	covid
1 and 1	160	171	170	covid
different result	43	31	28	covid
−1 and −1	7	9	43	japanese_credit
1 and 1	566	521	562	japanese_credit
different result	117	160	85	japanese_credit
−1 and −1	18	17	35	phishing
1 and 1	1492	1512	1530	phishing
different result	332	313	277	phishing
−1 and −1	0	2	5	SPECT
1 and 1	230	228	212	SPECT
different result	37	37	50	SPECT

**Table 6 entropy-23-00869-t006:** Data objects belonging to the car dataset classified as outliers by all algorithms: STIRR, ROCK, and K−modes assuming there is 10% of outliers in the set.

	Class	Buying	Maint	Doors	Persons	Lug_BOOT	Safety	ROCK Clusters	STIRR Weights	*K-modes* Clusters	ROCK Labels	STIRR Labels	*K-modes* Labels
7	vhigh	vhigh	2	2	big	med	unaccs	0	0.10832	48	−1	−1	−1
566	high	high	2	more	big	high	acc	0	0.043724	8	−1	−1	−1
970	med	vhigh	5more	more	big	med	acc	0	0.081686	4	−1	−1	−1

**Table 7 entropy-23-00869-t007:** Data objects belonging to the house votes dataset classified as outliers by all algorithms: STIRR, ROCK, and K−modes assuming there is 10% of outliers in the set.

	Political Party	A1	A2	A3	A4	A5	...	A16	ROCK Clusters	STIRR Weights	*K-modes* Clusters	ROCK Labels	STIRR Labels	*K-modes* Labels
276	republican	n	n	y	y	y	...	y	0	−0.047932	1	−1	−1	−1
315	democrat	n	n	n	n	n	...	n	0	−0.023109	1	−1	−1	−1
354	republican	y	n	y	y	n	...	y	0	−0.021921	4	−1	−1	−1
397	democrat	n	y	y	n	y	...	y	0	−0.042973	5	−1	−1	−1

**Table 8 entropy-23-00869-t008:** Data objects belonging to the *japanese credit* dataset classified as outliers by all algorithms: STIRR, ROCK, and K−modes assuming there there is 10% of outliers in the set.

	A1	A2	A3	A4	...	A16	ROCK Clusters	STIRR Weights	*K-modes* clusters	ROCK Labels	STIRR Labels	*K-modes* Labels
194	b	(33.7, 40.35]	(2.8, 5.6]	y	...	+	0	−0.000756	3	−1	−1	−1
360	a	(27.05, 33.7]	(2.8, 5.6]	u	...	−	0	−0.001686	6	−1	−1	−1
552	b	(33.7, 40.35]	(14.0, 16.8]	u	...	+	0	−0.007381	7	−1	−1	−1

**Table 9 entropy-23-00869-t009:** Data objects belonging to the *spect* dataset classified as outliers by all algorithms: STIRR, ROCK, and K−modes assuming there there is 10% of outliers in the set.

	F0	F1	F2	F3	...	F22	ROCK Clusters	STIRR Weights	*K-modes* Clusters	ROCK labels	STIRR Labels	*K-modes* Labels
86	1	1	0	0	...	1	0	0.004665	0	−1	−1	−1

**Table 10 entropy-23-00869-t010:** Data objects belonging to the *alpha bank* dataset classified as outliers by all algorithms: STIRR, ROCK, and K−modes assuming there there is 10% of outliers in the set.

	Age	Job	Marital_Status	Education	Default_Credit	Housing_Loan	...	ROCK Clusters	STIRR Weights	*K-modes* Clusters	ROCK Labels	STIRR Labels	*K-modes* Labels
0	(32.6, 40.4]	housem	sin	Sec_Ed	no	yes	...	0	−0.022802	9	−1	−1	−1
280	(24.8, 32.6]	tech	sin	Tert_Ed	no	no	...	0	−0.010348	0	−1	−1	−1
558	(71.6, 79.4]	retired	div	Prof_Ed	no	yes	...	0	0.021198	37	−1	−1	−1
569	(32.6, 40.4]	serv	mar	Prof_Ed	no	yes	...	0	−0.004888	16	−1	−1	−1
678	(32.6, 40.4]	serv	div	Prim_Ed	no	no	...	0	0.006195	16	−1	−1	−1
1573	(24.8, 32.6]	admin.	sin	Tert_Ed	no	no	...	0	0.003411	0	−1	−1	−1
1988	(56.0. 63.8]	unemp	mar	Prim_Ed	no	no	...	0	0.013755	56	−1	−1	−1
2127	(24.8, 32.6]	admin.	sin	Prof_Ed	no	no	...	0	0.018941	53	−1	−1	−1
2178	(32.6, 40.4]	housem	mar	Prim_Ed	no	yes	...	0	−0.008857	20	−1	−1	−1
2214	(40.4, 48.2]	unemp	mar	Prof_Ed	no	no	...	0	0.002494	13	−1	−1	−1
2322	(24.8, 32.6]	self-emp	div	Tert_Ed	no	no	...	0	−0.010603	0	−1	−1	−1
2397	(32.6, 40.4]	tech	div	Tert_Ed	no	yes	...	0	0.002534	6	−1	−1	−1
2514	(32.6, 40.4]	tech	div	Sec_Ed	no	yes	...	0	0.010159	16	−1	−1	−1

**Table 11 entropy-23-00869-t011:** Data objects belonging to the *phishing* dataset classified as outliers by all algorithms: STIRR, ROCK, and K−modes assuming there there is 10% of outliers in the set.

	UsingIP	LongURL	ShortURL	Symbol@	...	Class	ROCK Clusters	STIRR Weights	*K-modes* Clusters	ROCK Labels	STIRR Labels	*K-modes* Labels
260	−1	−1	−1	1	...	−1	0	0.028377	0	−1	−1	−1
616	−1	−1	1	−1	...	−1	0	−0.02149	2	−1	−1	−1
1382	−1	1	−1	1	...	1	0	−0.021263	0	−1	−1	−1

## Data Availability

Not applicable.

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
