# Peer review of "Qualitative Data Clustering to Detect Outliers"

_entropy, 2021, doi:10.3390/e23070869_

Round 1
Reviewer 1 Report
The paper presents a study on application of clustering algorithms for categorical data to outlier analysis.
The work seems original, however its novelty is limited. The paper has a proper structure and a significant list of proper references.
I have the following comments to the work.
1.
The article contains a lot of content, from which it is often difficult to extract the essence - it would be good to make it more concise.
E.g. the introduction takes three pages and it is missing a clear contribution statement, and the goal of the work was included in one sentence at the end of this section. Moreover, the motivation for this work is distributed between sections 1. Introduction and 2. State of art. A clear statement of the motivation, goal and contribution of this work should be presented in the Introduction section.
2.
The three clustering algorithms were selected to perform analysis, however, there is no justification why these three methods were selected. There are several other methods that could be utilised when an appropriate similarity/distance measure was used.
3.
The clustering algorithms analysed in the study are not new and in my opinion there is no need to present them again. I think it is enough to refer to the original works introducing these methods, unless any specific changes were applied.
In my opinion this part should be removed but, by the way, in Definition 1 the notation of x_1 for X and x is the same whereas these are an object and an attribute and they have to be presented differently.
4.
As a result of comparing the analysed algorithms, I would expect some conclusions supporting the thesis that motivated the experiments, e.g. the conclusions regarding quality and/or usefulness of the methods. It would be interesting to find hints when to use which algorithm, e.g. which algorithm for what kind of data or what kind of outliers works better. Conclusions regarding the time of analysis for each method come as no surprise, knowing their computational complexities.
5.
Fig. 2 - any descriptions on the figure should be in English.
6.
"Most models based on categorical data are designed to be rules, therefore the presence of noise in the set can significantly disturb the learning process." - this sentence is unclear, therefore, the transition to the topic of rules is unclear
7.
Typos and language errors:
- presents presents
- gdzie
- mods
- It's
Reviewer 2 Report
Paper presents an experimental comparison of three categorical data clustering algorithms, namely: K-modes, STIRR and ROCK, in the detection of outliers.
Major concerns:
- Authors considered only four databases, so testing cannot be considered exhaustive. Authors must include more databases and perform an statistical test so as to provide results that are less subjective.
- Databases used for comparison purposes are generic; that is, they were not designed and reported to be used for outlier detection. So experimental results lack of a ground truth against which one can measure the performance of the methods in outlier detection. However, authors state, in p.2 that "Hence, in this paper, we undertook the task of analyzing available solutions in terms of detecting anomalies in qualitative data sets. We have focused in particular on describing data clustering methods that are used to detect variances in qualitative sets, as they are the most effective methods of detecting anomalies that do not need to perform calculations on numerical domains so far." As stands, this statement is misleading with respect to paper content. Authors are so encouraged to use datasets that have stemed in the anomaly detection context, such as masquerade detection (eg WUIL [1]), fraud detection, etc.
- The description of the methods is obscure and sloppy. Language and notation incurs in overloading, without the authors making any notice. Further, there are expresions that do not type check: what is the meaning of x_1\cap x_2, when x_i (i en {1,2}) is a tuple. Sometimes, authors introduce terms that later on are no longer used (see step 3 of Algorithm 1). What is $m$ there, anyway?
- In the description of algorithms, authors use terms which should be formally defined. For example, what is the meaning of uniqueness of a value that a variable may take?
- In p.5, authors state that: "Most outlier detection algorithms focus on numerical attributes whose natural geometric properties can be used to define the distance function between points. Currently, there has been an increased demand for processing data of a qualitative or categorical nature, the attribute values of which do not have a natural ordering. Calculating statistical values based on this type of data is difficult and in some cases impossible to perform." This referee disagrees with this statement, for it is always possible to use a (dis)similarity measure with which we can have a means for object comparison, making it possible to use other algorithms (and of course the algorithms considered in the paper.)
- In p.2, authors state: "Numbers such as a telephone number, social security number, etc. are considered qualitative data as they are categorical and unique to one person." Data consists of a collection of objects, each of which is represented as a tuple of data items. Data mining is trivial where all these data items' values are unique. I suspect you meant something different here.
- Make sure any technical term you use is properly introduced earlier. Examples: cluster combination, number of points, etc.
MINOR CONCERNS:
- Replace "gdzie" with "where"
- Verify paper does not contain other typos or grammatical errors. For example, in p.3 you wrote: "The authors of [22] presents ...", which has a number agreement grammatical error.
- Verify citation usage, for the one in the paper does not look scholastic.
- Verify references are not obscure; example: in p. 6, we have "The article analyzes ..." what does "the article" refer to? This article? Someone else's article?
- "outliers detection" --> "outlier detection"
- "Neither of the algorithms" --> "None of these algorithms"
- Be consistent: sometimes you write "data set" and some other times "dataset". Stick to one term
[1] J. Benito Camiña, C. Hernández-Gracidas, R. Monroy and L. Trejo. The Windows-Users and -Intruder simulations Logs Dataset (WUIL): An Experimental Framework for Masquerade Detection Mechanisms. Expert Systems with Applications, Methods and Applications of Artificial and Computational Intelligence. 41(3):919 - 930. Februray 15, 2014.
[2] E.-J. Rivera-Ríos, M.-A. Medina-Pérez, M.-S. Lazo-Cortés, R. Monroy. Learning-Based Dissimilarity for clustering categorical data. Applied Siences. 11(8):3509, pp. 1-17.
Round 2
Reviewer 1 Report
Some of my comments were ignored - neither corrections were made in the paper, nor were any discussions included in the response to the review.
There is still no clear statement of the motivation, goal and contribution of this work in the Introduction section.
I still do not see any point in rewriting methods (and adding confusion to the definitions) that have already been published.
The introduction to section 6.5 is not related to the experimental results, but would fit with the rationale for selecting the clustering algorithms that is still not added to the article.
Author Response
Dear Reviewer,
Thank you for all your valuable comments and tips. They certainly helped improve the paper and allowed us to look at our research from a different perspective. We hope we were able to respond to every comment and advice. We took into account all of them and added the necessary clarifications to the text of the manuscript.

Reviewer 2 Report
Authors have gone through all my observations and so I recommend paper acceptance.
Author Response

(The authors gave the same response as above.)

Round 3
Reviewer 1 Report
My previous comments were addressed in the work.